# The Association of Beta-Blocker Use and Bone Mineral Density Level in Hemodialysis Patients: A Cross-Sectional Study

**DOI:** 10.3390/medicina59010129

**Published:** 2023-01-09

**Authors:** Suthiya Anumas, Saranya Thitisuriyarax, Pichaya Tantiyavarong, Waroot Pholsawatchai, Pattharawin Pattharanitima

**Affiliations:** 1Chulabhorn International College of Medicine, Thammasat University, Pathumthani 12120, Thailand; 2Division of Nephrology, Department of Medicine, Faculty of Medicine, Thammasat University, Pathumthani 12120, Thailand; 3Department of Clinical Epidemiology, Faculty of Medicine, Thammasat University, Pathumthani 12120, Thailand

**Keywords:** beta-blockers, bone mineral density, hemodialysis patients

## Abstract

*Background and Objectives*: Osteoporosis results in increasing morbidity and mortality in hemodialysis patients. The medication for treatment has been limited. There is evidence that beta-blockers could increase bone mineral density (BMD) and reduce the risk of fracture in non-dialysis patients, however, a study in hemodialysis patients has not been conducted. This study aims to determine the association between beta-blocker use and bone mineral density level in hemodialysis patients. *Materials and Methods*: We conducted a cross-sectional study in hemodialysis patients at Thammasat University Hospital from January 2018 to December 2020. A patient receiving a beta-blocker ≥ 20 weeks was defined as a beta-blocker user. The association between beta-blocker use and BMD levels was determined by univariate and multivariate linear regression analysis. *Results*: Of the 128 patients receiving hemodialysis, 71 were beta-blocker users and 57 were non-beta-blocker users (control group). The incidence of osteoporosis in hemodialysis patients was 50%. There was no significant difference in the median BMD between the control and the beta-blocker groups of the lumbar spine (0.93 vs. 0.91, *p* = 0.88), femoral neck (0.59 vs. 0.57, *p* = 0.21), total hip (0.73 vs. 0.70, *p* = 0.38), and 1/3 radius (0.68 vs. 0.64, *p* = 0.40). The univariate and multivariate linear regression analyses showed that the beta-blocker used was not associated with BMD. In the subgroup analysis, the beta-1 selective blocker used was associated with lower BMD of the femoral neck but not within the total spine, total hip, and 1/3 radius. The multivariate logistic regression showed that the factors of age ≥ 65 years (aOR 3.31 (1.25–8.80), *p* = 0.02), female sex (aOR 4.13 (1.68–10.14), *p* = 0.002), lower BMI (aOR 0.89 (0.81–0.98), *p* = 0.02), and ALP > 120 U/L (aOR 3.88 (1.33–11.32), *p* = 0.01) were independently associated with osteoporosis in hemodialysis patients. *Conclusions*: In hemodialysis patients, beta-blocker use was not associated with BMD levels, however a beta-1 selective blocker used was associated with lower BMD in the femoral neck.

## 1. Introduction

Osteoporosis from chronic kidney disease-mineral and bone disease (CKD-MBD) can be defined as impaired bone health, causing a higher risk of bone fracture, and is associated with higher morbidity and mortality [1,2]. For bone strength evaluation, the Kidney Disease Improving Global Outcome (KDIGO) CKD-MBD guideline from 2017 recommends bone mineral density (BMD) testing in patients with CKD G3a–G5D with evidence of CKD-MBD and/or risk factors for osteoporosis to assess fracture risk if results impact treatment decisions [3]. However, the current medications that treat osteoporosis in hemodialysis patients (CKD G5D) are of limited availability. Drugs with possible benefits, including antiresorptive agents and anabolic hormones, are expensive and have side effects that require close monitoring during usage.

Beta-blockers are antihypertensive drugs that are widely used in hemodialysis patients. Besides the benefits of controlling blood pressure and cardiovascular disease, many studies have shown the benefits of beta-blockers in improving BMD and lowering the risk of bone fractures in the general population [4,5,6,7,8,9,10].

Studies in animal models have shown that bone formation is regulated by the leptin–sympathetic nervous system pathway. The elevation in leptin stimulates beta-adrenergic and sympathetic responses. The increase in sympathetic activity then induces osteoblasts to release the receptor activator of nuclear factor kappa-B ligand (RANKL), which signals the activation of osteoclasts and bone resorption. Beta-adrenergic receptor blocker plays the role of an antisympathetic agent which reduces intracerebral leptin [11,12,13] and directly binds to the beta-2 adrenergic receptor on the osteoblasts, resulting in the inhibition of RANKL and reduction in osteoclast activity [14,15]. These changes in bone remodeling may explain the overall effect of beta-blockers on the reduction in bone resorption and the increase in bone mass. However, there has been no study conducted in hemodialysis patients. Therefore, the aim of this study is to determine the association between beta-blocker use and BMD in patients receiving hemodialysis. 

## 2. Materials and Methods

We conducted a single-center, cross-sectional study at Thammasat University Hospital in patients with adequate hemodialysis from 1 January 2018 to 31 October 2022. 

We retrospectively reviewed all medical records using a standard case report form. Data were collected, including demographic characteristics, comorbidities, dialysis vintage, history of alcohol consumption and smoking, current medications, mean laboratory data within 1 year before performing BMD, and BMD data. The patients with ages of 18 or more who received adequate hemodialysis for at least 90 days were included. The beta-blocker users were patients who were currently using and had received a beta-blocker for at least 20 weeks. Patients who had incomplete medical records, post-parathyroidectomy, were receiving anti-osteoporotic treatment within 6 months, were receiving hormone replacement therapy, were taking calcimimetic drugs or immunosuppressive agents, and a history of malignancy or bone metastasis were excluded. 

The primary objective was to assess the association between beta-blocker usage and BMD of the femoral neck, total hip, and 1/3 radius of hemodialysis patients. The secondary objectives were to determine the incidence of osteoporosis in hemodialysis patients and to examine the factors influencing osteoporosis in hemodialysis patients which were diagnosed by a T-score ≤ −2.5 of the total spine, femoral neck, total hip, or 1/3 radius.

Categorical variables were reported as a frequency and percentage, and numerical data were reported as a mean with a standard deviation (SD) or a median with an interquartile range (IQR) where appropriate. Comparison of the categorical and continuous data between the two groups was conducted by Fisher’s exact test and Wilcoxon’s rank sum test, respectively. Univariate and multivariate linear regression analyses were performed to determine the factors associated with BMD. Univariate and multivariate logistic regression was performed to determine the risk factors of osteoporosis. The covariates with *p* ≤ 0.1 from the univariate model were included in the multivariate model. A statistical significance was considered as *p* < 0.05. Data were analyzed using STATA version 16.

## 3. Results

From 1 January 2018 to 31 October 2022, 128 hemodialysis patients were included. A total of 57 patients were non-beta-blocker users (control group) and 71 patients were beta-blocker users. Of them, 53.1% of patients were female with a mean (SD) age of 65 (15) years and median (IQR) dialysis vintage of 32 (14–64.5) months. The baseline characteristics were not significantly different between the two groups (Table 1). Among the beta-blocker users, 23 of them used beta-1 selective blockers and 48 used non-selective beta-blockers. The baseline characteristics were not significantly different, except in serum bicarbonate levels, which were slightly higher in the non-selective beta-blocker users (25 (24.8–48.4) vs. 24 (23.3–24.7) mEq/L, *p* = 0.03) (Appendix A).

There was no difference in median BMD between the beta-blocker and control groups, including in total lumbar spine BMD (0.91 vs. 0.93, *p* = 0.88), femoral neck BMD (0.57 vs. 0.59, *p* = 0.21), total hip BMD (0.70 vs. 0.73, *p* = 0.38), and 1/3 radius BMD (0.68 vs. 0.64, *p* = 0.40), respectively (Figure 1, Appendix A).

From univariate linear regression analysis, the factors associated with BMD of the femoral neck were female, age ≥ 65 years, calcium < 8.5 mg/dL, and ALP > 120 U/L. Multivariate analysis showed that only the factors of age ≥ 65 years and female were associated with BMD of the femoral neck in Models 1 and 2. Beta-blocker use was not associated with BMD of the femoral neck in any models (Table 2). Beta-1 selective blocker use was significantly associated with BMD of the femoral neck in Model 1 (coefficient −0.17 (−0.33 to −0.005), *p* = 0.04), but not Model 2 (coefficient −0.16 (−0.33 to 0.003), *p* = 0.054) (Appendix A).

From univariate linear regression analysis, the factors associated with BMD of the total hip were female, age ≥ 65 years, lower BMI, and ALP > 120 U/L. Multivariate analysis showed that the factors of age ≥ 65 years, female, BMI, and ALP > 120 U/L were independently associated with BMD of the total hip in Models 1 and 2. Beta-blocker use was not associated with BMD of the total hip in any models (Table 3).

From univariate linear regression analysis, the factors associated with BMD of the 1/3 radius were female, age ≥ 65 years, former smoking, dialysis vintage, PTH < 120 pg/mL, lower BMI, and ALP > 120 U/L. Multivariate analysis showed that the factors of age ≥ 65 years, female, and ALP > 120 U/L were independently associated with BMD of the 1/3 radius in Models 1 and 2. Beta-blocker use was not associated with BMD of the 1/3 radius in any models (Table 4).

From univariate linear regression analysis, the factors associated with BMD of the total spine were female, BMI, ischemic stroke, gout, ACEi use, PTH > 585 pg/mL, and ALP > 120 U/L. Multivariate analysis showed that the factors of female, BMI, and gout were independently associated with BMD of the total spine in Models 1 and 2. Beta-blocker use was not associated with BMD of the total spine in any models (Table 5).

The incidence of osteoporosis was 50% among all patients. When the on-site T-score ≤ −2.5 was investigated, the two most common sites were the femoral neck (35.94%) and 1/3 radius (35.16%) (Appendix A). However, no difference in the incidence between the beta-blocker users and the control group was found. 

From univariate logistic regression analysis, the risk factors of osteoporosis were female, age ≥ 65 years, lower BMI, and ALP > 120 U/L. Multivariate analysis showed that the factors of age ≥ 65 years (aOR 3.31 (1.25–8.80), *p* = 0.02), female (aOR 4.13 (1.68–10.14), *p* = 0.002), lower BMI (aOR 0.89 (0.81–0.98), *p* = 0.02), and ALP > 120 U/L (aOR 3.88 (1.33–11.32), *p* = 0.01) were independently associated with osteoporosis (Table 6).

## 4. Discussion

This is the first study to investigate the association between beta-blocker usage and BMD in hemodialysis patients. The study demonstrated that beta-blocker use was not associated with BMD levels, however in the subgroup analysis, beta-1 selective blocker use was associated with a lower BMD of the femoral neck. The prevalence of osteoporosis in hemodialysis patients was 50% and the risk factors for osteoporosis were age ≥ 65 years, female, lower BMI, and ALP over 120 U/L. 

The association of beta-blocker use and BMD was found in non-hemodialysis patients, however the results remained inconsistent. Seven studies showed beta-blocker use was associated with higher BMD [4,5,6,7,8,9,10], but not in three other studies [16,17,18]. Compared to our study, our populations had a lower median femoral neck BMD than the population from previous positive studies (0.57 vs. 0.7–0.9) which might explain why the benefits of beta-blockers were not present in our study. Moreover, the most common mechanism of renal osteodystrophy in hemodialysis patients is not a high bone turnover but a low bone turnover process which has a prevalence of 56.1% [1]. In our study, only 16.41% of patients with a high level of both bone turnover markers (high PTH and ALP) possibly had a high turnover process. Only 2.34% of our patients had both low bone turnover markers (low PTH and ALP) and 82.46% had an undetermined process which needed a bone biopsy to prove. Since beta-blocker decreases bone remodeling via antisympathetic activity [11,12,13], the benefits of beta-blockers should be found only in the patients with high bone turnover process, but not in patients with a low bone turnover or undetermined bone turnover process (Appendix A). In our study, there was a high proportion of patients who possibly had a low bone turnover and undetermined bone turnover process; therefore, beta-blocker use did not associate with a higher BMD and even seemed to worsen it. 

One meta-analysis [19] demonstrated that beta-1 selective blocker provides the benefit of reduced risk of fracture compared to non-selective beta-blocker which could be mediated systemically, unlike the direct effect on the beta-2 receptor of osteoclasts and osteoblasts. Nevertheless, our study demonstrated that selective beta-1 blocker was associated with lower BMD of the femoral neck. Since the BMD of the total spine might be falsely high from vascular calcification [20], which is the result of CKD-MBD, using BMD of the total spine to diagnose osteoporosis might be falsely negative. Thus, we focused on the BMD of the femoral neck, total hip, and 1/3 radius as the primary outcomes.

Compared with the study by Slouma et al. [21], the incidence of osteoporosis in hemodialysis patients in our study was higher, which might be explained by the higher mean age of our population (65 ± 15 vs. 53 ± 14). In contrast, our study found that the femoral neck and 1/3 radius were the most common sites of osteoporosis, while it was the total hip for the study by Slouma et al.

In concordance with our findings, we found the risk factors of osteoporosis were age ≥ 65 years, female, lower BMI, and ALP > 120 U/L. The study by Slouma et al. also found that age was a risk factor for osteoporosis; however this was not consistent with another study [1].

There were a few limitations in our study. First, the relatively small sample size of the population might result in a modest power of the study. Second, due to the nature of a cross-sectional study, the results could not indicate a causation. Third, the definite duration and dose of beta-blocker use were not available and they might affect the benefit of beta-blocker use on BMD.

## 5. Conclusions

There was no association between beta-blocker use and BMD levels in hemodialysis patients; however, the subgroup analysis showed beta-1 selective blocker was associated with a lower BMD of the femoral neck. The risk factors for osteoporosis in hemodialysis patients were age ≥ 65 years, sex, low BMD, and ALP over 120 U/L. Further studies to evaluate the effect of beta-blocker on BMD with a larger sample size and randomized controlled design in hemodialysis patients should be considered.

## Figures and Tables

**Figure 1 medicina-59-00129-f001:**
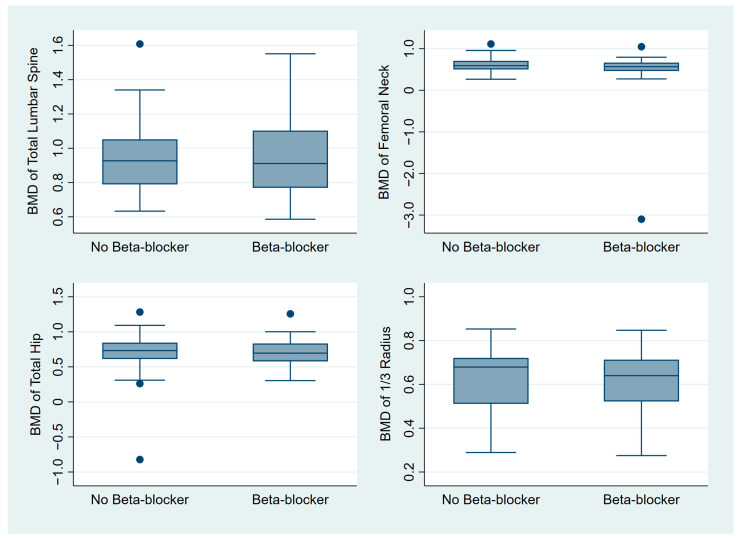
Box plot graphs comparing the BMD between the beta-blocker and control groups. Abbreviation: BMD, bone mineral density.

**Table 1 medicina-59-00129-t001:** Baseline characteristics of patients.

Characteristics	Total (N = 128)	Non-Beta-Blocker Users (N = 57)	Beta-Blocker Users (N = 71)	*p*-Value
Age, years, mean (SD)	65 (15)	68 (15)	64 (16)	0.13
Female, *n* (%)	68 (53.1)	26 (45.6)	42 (59.1)	0.15
BMI, kg/m^2^, mean (SD)	24.2 (4.8)	23.7 (4.5)	24.6 (5.1)	0.35
Alcohol, *n* (%)				0.95
• Never	91 (71.1)	40 (70.2)	51 (71.8)	
• Current	9 (7.0)	4 (7.0)	5 (7.0)	
• Former	28 (21.9)	13 (22.8)	15 (21.1)	
Former Smoking, *n* (%)	15 (11.7)	5 (8.8)	10 (14.1)	0.42
Underlying disease, *n* (%)				
• Dyslipidemia	76 (59.8)	36 (63.2)	40 (56.3)	0.47
• Ischemic stroke	8 (6.2)	2 (3.5)	6 (8.5)	0.30
• Ischemic heart disease	24 (18.7)	7 (12.3)	17 (23.9)	0.11
• Gout	19 (14.8)	9 (15.8)	10 (14.1)	0.81
• Hypertension	119 (93.0)	53 (93.0)	66 (93.0)	1.00
• Diabetes mellitus	69 (54.0)	34 (59.6)	35 (49.3)	0.29
• Atrial fibrillation	13 (10.2)	6 (10.5)	7 (9.9)	1.00
Dialysis vintage, months, median (IQR)	32.0 (14.0–64.5)	32.0 (11.0–63.0)	33.0 (20.0–65.0)	0.30
Beta-blockers type, *n* (%)				
• No beta-blocker	57 (44.5)	57 (100.0)	0	
• Non-selective beta-blockers	48 (37.5)	0	48 (67.6)	
• Beta-1 selective blockers	23 (17.94)	0	23 (32.4)	
Antihypertensive drugs, *n* (%)				
• ACEi	5 (3.9)	2 (3.5)	3 (4.2)	1.00
• ARB	43 (33.6)	14 (24.6)	29 (40.8)	0.06
• Alpha-1 blocker	38 (29.7)	19 (33.3)	19 (26.8)	0.42
• CCB	91 (71.1)	38 (66.7)	53 (74.6)	0.33
• Vasodilator	35 (27.3)	14 (24.6)	21 (29.6)	0.56
Statin, *n* (%)	86 (67.2)	38 (66.7)	48 (67.6)	1.00
Non-calcium-based phosphate binder, *n* (%)	24 (18.7)	14 (24.6)	10 (14.1)	0.17
Active vitamin D, *n* (%)	61 (47.7)	28 (49.1)	33 (46.5)	0.90
Laboratory				
• Calcium, mg/dL, median (IQR)	9.0 (8.5–9.5)	9.0 (8.5–9.5)	8.9 (8.4–9.4)	0.64
• Phosphate, mg/dL, median (IQR)	4.6 (3.8–5.3)	4.4 (3.6–5.3)	4.6 (3.8–5.3)	0.47
• PTH, pg/mL, median (IQR)	477.2 (331.0–647.8)	515.6 (341.2–688.0)	457.0 (311.0–632.0)	0.26
• PTH level, *n* (%)				0.94
PTH < 120 pg/mL	3 (2.34%)	1 (1.75)	2 (2.82)	
PTH = 120–585 pg/mL	82 (64.06%)	36 (63.16)	46 (62.79)	
PTH > 585 pg/mL	43 (33.59)	20	23 (32.39)	
• ALP, U/L, median (IQR)	92.0 (67.3–126.0)	95 (70.0–126.0)	90 (66.0–127.0)	0.52
• ALP level, *n* (%)				1.00
ALP ≤ 120 U/L	90 (70.31)	40 (70.18)	50 (70.42)	
ALP > 120 U/L	38 (29.69)	17 (29.82)	21 (29.58)	
• Vitamin D, ng/mL, median (IQR)	35.9 (26.4–44.3)	34.9 (25.5–42.7)	36.8 (28.3–49.1)	0.19
• Bicarbonate, mEq/L, median (IQR)	24.4 (23.6–25.6)	24.4 (23.3–25.5)	24.4 (23.7–26.0)	0.86
• nPCR, g/kg/day, median (IQR)	1.10 (0.98–1.32)	1.10 (0.99–1.30)	1.10 (0.88–1.32)	0.21
• URR, %, median (IQR)	80.3 (77.8–83.4)	80.1 (77.2–82.4)	80.6 (78.2–83.7)	0.12
• Albumin, g/dL, median (IQR)	3.74 (3.52–3.90)	3.74 (3.52–3.90)	3.63 (3.52–3.90)	0.65
Bone turnover *, *n* (%)				1.00
• High turnover	21 (16.41)	9 (15.79)	12 (16.90)	
• Low turnover	3 (2.34)	1 (1.75)	2 (2.82)	
• Undetermined	47 (82.46)	47 (82.46)	57 (80.28)	

Abbreviations: ACEi, angiotensin-converting enzyme inhibitor; ALP, alkaline phosphatase; ARB, angiotensin II receptor blocker; BMI, body mass index; CCB, calcium channel blockers; nPCR, normalized protein catabolic rate; PTH, parathyroid hormone; URR, urea reduction ratio. * High turnover defined as high PTH and ALP, low turnover defined as low PTH and ALP, undetermined defined as high PTH with low ALP or low PTH with high ALP.

**Table 2 medicina-59-00129-t002:** Association of beta-blocker use and BMD of the femoral neck.

Factors	Univariate		Multivariate			
			Model 1		Model 2	
	Coefficient (95% CI)	*p*-Value	Coefficient (95% CI)	*p*-Value	Coefficient (95% CI)	*p*-Value
Age ≥ 65 years	−0.14 (−0.27 to −0.02)	0.02	−0.12 (−0.24 to −0.004)	0.04	−0.13 (−0.25 to −0.07)	0.04
Female	−0.17 (−0.29 to −0.04)	0.01	−0.15 (−0.27 to −0.03)	0.01	−0.15 (−0.27 to−0.03)	0.01
BMI	0.01 (−0.002 to 0.02)	0.10	0.01 (−0.002 to 0.02)	0.09	0.01 (−0.002 to 0.22)	0.11
Alcohol	−0.05 (−0.12 to 0.03)	0.21				
Former smoking	0.05 (−0.05 to 0.15)	0.33				
Dyslipidemia	0.10 (−0.02 to 0.23)	0.11				
Ischemic stroke	0.05 (−0.21 to 0.32)	0.68				
Ischemic heart disease	−0.16 (−0.31 to 0.003)	0.06	−0.12 (−0.27 to 0.04)	0.14	−0.11 (−0.27 to 0.04)	0.16
Gout	0.09 (−0.09 to 0.27)	0.32				
Hypertension	−0.11 (−0.36 to 0.13)	0.36				
Diabetes mellitus	0.05 (−0.08 to 0.17)	0.46				
Atrial fibrillation	−0.01 (−0.22 to 0.20)	0.90				
Dialysis vintage (months)	−0.0004 (−0.002 to 0.001)	0.63				
Beta-blocker use	−0.09 (−0.21 to 0.04)	0.18	−0.06 (−0.18 to 0.05)	0.28	−0.06 (−0.18 to 0.06)	0.30
Statin	−0.05 (−0.19 to 0.08)	0.42				
ACEi	−0.085 (−0.41 to 0.24)	0.60				
ARB	0.06 (−0.07 to 0.19)	0.39				
Non-calcium-based phosphate binder	0.10 (−0.06 to 0.26)	0.23			0.03 (−0.12 to 0.19)	0.69
Active vitamin D	−0.06 (−0.18 to 0.07)	0.38			−0.05 (−0.14 to 0.06)	0.36
Calcium < 8.5 mg/dL	−0.15 (−0.29 to −0.01)	0.03	−0.11 (−0.24 to 0.03)	0.11	−0.11 (−0.25 to 0.03)	0.12
Phosphate						
• Normal phosphate	1					
• Low phosphate (<3 mg/dL)	−0.02 (−0.22 to 0.17)	0.80				
• High phosphate (>5 mg/dL)	−0.04 (−0.18 to 0.10)	0.55				
PTH						
• PTH 120–585 pg/mL	1					
• PTH < 120 pg/mL	0.03 (−0.39 to 0.45)	0.90				
• PTH > 585 pg/mL	0.01 (−0.12 to 0.14)	0.89				
Vitamin D	0.0008 (−0.003 to 0.005)	0.71				
Bicarbonate	0.02 (−0.01 to 0.05)	0.18				
ALP > 120 U/L	−0.21 (−0.34 to −0.08)	0.002	−0.11 (−0.25 to 0.02)	0.10	−0.11 (−0.25 to 0.03)	0.12
Albumin < 3.5 g/dL	−0.02 (−0.18 to 0.14)	0.81				

Model 1: Adjusted covariates with *p* ≤ 0.1 from the univariate model. Model 2: Model 1 + non-calcium-based phosphate binder and active vitamin D supplementation. Abbreviations: ACEi, angiotensin-converting enzyme inhibitor; ALP, alkaline phosphatase; ARB, angiotensin II receptor blocker; BMI, body mass index; PTH, parathyroid hormone.

**Table 3 medicina-59-00129-t003:** Association of beta-blocker use and BMD of the total hip.

Factors	Univariate		Multivariate			
			Model 1		Model 2	
	Coefficient (95% CI)	*p*-Value	Coefficient (95% CI)	*p*-Value	Coefficient (95% CI)	*p*-Value
Age ≥ 65 years	−0.11 (−0.19 to −0.03)	0.01	−0.08 (−0.15 to −0.01)	0.03	−0.08 (−0.15 to −0.01)	0.03
Female	−0.14 (−0.22 to −0.06)	<0.001	−0.14 (−0.21 to −0.07)	<0.001	−0.14 (−0.21 to−0.07)	<0.001
BMI	0.01 (0.007 to 0.02)	<0.001	0.01 (0.008 to 0.02)	<0.001	0.01 (0.008 to 0.02)	<0.001
Alcohol	−0.01 (−0.04 to 0.06)	0.59				
Former smoking	0.04 (−0.02 to 0.10)	0.22				
Dyslipidemia	0.05 (−0.03 to 0.14)	0.21				
Ischemic stroke	0.03 (−0.14 to 0.20)	0.73				
Ischemic heart disease	0.01 (−0.09 to 0.11)	0.84				
Gout	0.02 (−0.10 to 0.13)	0.78				
Hypertension	−0.06 (−0.22 to 0.10)	0.47				
Diabetes mellitus	0.01 (−0.07 to 0.10)	0.76				
Atrial fibrillation	−0.03 (−0.17 to 0.11)	0.66				
Dialysis vintage (months)	−0.0006 (−0.002 to 0.0004)	0.23				
Beta-blocker use	−0.006 (−0.09 to 0.08)	0.88	−0.0004 (−0.07 to 0.07)	0.99	−0.0004 (−0.07 to 0.07)	0.99
Statin	0.03 (−0.06 to 0.12)	0.48				
ACEi	−0.14 (−0.35 to 0.07)	0.18				
ARB	−0.04 (−0.13 to 0.04)	0.32				
Non-calcium-based phosphate binder	0.09 (−0.02 to 0.19)	0.10	0.05 (−0.05 to 0.14)	0.33	0.05 (−0.05 to 0.14)	0.33
Active vitamin D	−0.0003 (−0.08 to 0.08)	0.99			0.002 (−0.07 to 0.07)	0.95
Calcium < 8.5 mg/dL	−0.06 (−0.15 to 0.03)	0.21				
Phosphate						
• Normal phosphate	1					
• Low phosphate (<3 mg/dL)	−0.03 (−0.16 to 0.09)	0.61				
• High phosphate (>5 mg/dL)	0.05 (−0.04 to 0.14)	0.26				
PTH						
• PTH 120–585 pg/mL	1					
• PTH < 120 pg/mL	0.06 (−0.22 to 0.33)	0.68				
• PTH > 585 pg/mL	0.03 (−0.12 to 0.06)	0.51				
Vitamin D	−0.00002 (−0.003 to 0.003)	0.99				
Bicarbonate	0.0007 (−0.02 to 0.02)	0.94				
ALP > 120 U/L	−0.13 (−0.22 to −0.05)	0.002	−0.08 (−0.16 to −0.0005)	0.048	−0.08 (−0.16 to −0.0002)	0.049
Albumin < 3.5 g/dL	−0.07 (−0.18 to 0.03)	0.16				

Model 1: Adjusted covariates with *p* ≤ 0.1 from the univariate model. Model 2: Model 1 + non-calcium-based phosphate binder and active vitamin D supplementation. Abbreviations: ACEi, angiotensin-converting enzyme inhibitor; ALP, alkaline phosphatase; ARB, angiotensin II receptor blocker; BMI, body mass index; PTH, parathyroid hormone.

**Table 4 medicina-59-00129-t004:** Association of beta-blocker use and BMD of the 1/3 radius.

Factors	Univariate		Multivariate			
			Model 1		Model 2	
	Coefficient (95% CI)	*p*-Value	Coefficient (95% CI)	*p*-Value	Coefficient (95% CI)	*p*-Value
Age ≥ 65 years	−0.06 (−0.11 to −0.01)	0.01	−0.06 (−0.10 to −0.02)	0.003	−0.06 (−0.10 to −0.02)	0.003
Female	−0.17 (−0.21 to −0.13)	<0.001	−0.14 (−0.18 to −0.10)	<0.001	−0.14 (−0.18 to −0.10)	<0.001
BMI	0.003 (−0.002 to 0.01)	0.22				
Alcohol	0.01 (−0.02 to 0.04)	0.38				
Former smoking	0.05 (0.02 to 0.09)	0.01	0.005 (−0.02 to 0.03)	0.75	0.006 (−0.02 to 0.04)	0.71
Dyslipidemia	0.002 (−0.05 to 0.05)	0.94				
Ischemic stroke	−0.004 (−0.10 to 0.10)	0.93				
Ischemic heart disease	0.02 (−0.04 to 0.08)	0.49				
Gout	0.06 (−0.01 to 0.13)	0.07	−0.01 (−0.07 to 0.04)	0.57	−0.01 (−0.07 to 0.04)	0.66
Hypertension	−0.05 (−0.14 to 0.05)	0.32				
Diabetes mellitus	−0.04 (−0.08 to 0.01)	0.15				
Atrial fibrillation	−0.02 (−0.10 to 0.06)	0.68				
Dialysis vintage (months)	−0.0007 (−0.001 to −0.0001)	0.02	−0.0005 (−0.0009 to 0.00004)	0.052	−0.0004 (−0.0009 to 0.00005)	0.08
Beta-blocker use	0.01 (−0.06 to 0.04)	0.61	0.004 (−0.03 to 0.04)	0.81	0.005 (−0.03 to 0.04)	0.77
Statin	0.05 (−0.004 to 0.10)	0.07	0.04 (−0.004 to 0.07)	0.07	0.03 (−0.009 to 0.07)	0.13
ACEi	−0.11 (−0.24 to 0.01)	0.08	−0.06 (−0.15 to 0.03)	0.21	−0.06 (−0.15 to 0.04)	0.23
ARB	0.002 (−0.05 to 0.05)	0.94				
Non-calcium-based phosphate binder	0.05 (−0.1 to 0.11)	0.13			0.01 (−0.03 to 0.06)	0.56
Active vitamin D	−0.03 (−0.08 to 0.01)	0.20			−0.02 (−0.06 to 0.02)	0.27
Calcium < 8.5 mg/dL	0.002 (−0.05 to 0.06)	0.94				
Phosphate						
• Normal phosphate	1					
• Low phosphate (<3 mg/dL)	−0.06 (−0.14 to 0.01)	0.11				
• High phosphate (>5 mg/dL)	−0.01 (−0.06 to 0.04)	0.77				
PTH						
• PTH 120–585 pg/mL	1					
• PTH < 120 pg/mL	−0.16 (−0.32 to 0.004)	0.04	−0.10 (−0.22 to 0.2)	0.11	−0.10 (−0.22 to 0.02)	0.11
• PTH > 585 pg/mL	−0.05 (−0.10 to 0.0005)	0.052	−0.03 (−0.07 to 0.01)	0.17	−0.02 (−0.06 to 0.02)	0.38
Vitamin D	−0.0007 (−0.001 to 0.002)	0.39				
Bicarbonate	−0.004 (−0.02 to 0.008)	0.56				
ALP > 120 U/L	−0.10 (−0.15 to −0.05)	<0.001	−0.05 (−0.09 to −0.003)	0.04	−0.05 (−0.09 to −0.002)	0.04
Albumin < 3.5 g/dL	−0.05 (−0.11 to 0.007)	0.08			−0.04 (−0.09 to 0.01)	0.12

Model 1: Adjusted covariates with *p* ≤ 0.1 from the univariate model. Model 2: Model 1 + non-calcium-based phosphate binder and active vitamin D supplementation. Abbreviations: ACEi, angiotensin-converting enzyme inhibitor; ALP, alkaline phosphatase; ARB, angiotensin II receptor blocker; BMI, body mass index; PTH, parathyroid hormone.

**Table 5 medicina-59-00129-t005:** Association of beta-blocker use and BMD of the total spine.

Factors	Univariate		Multivariate			
			Model 1		Model 2	
	Coefficient (95% CI)	*p*-Value	Coefficient (95% CI)	*p*-Value	Coefficient (95% CI)	*p*-Value
Age ≥ 65 years	−0.39 (−1.02 to 0.23)	0.21				
Female	−1.26 (−1.84 to −0.67)	<0.001	−1.06 (−1.64 to −0.47)	<0.001	−1.04 (−1.63 to −0.45)	0.001
BMI	0.11 (0.05 to 0.17)	0.001	0.12 (0.06 to 0.18)	<0.001	0.12 (0.06 to 0.18)	<0.001
Alcohol	0.10 (−0.27 to 0.48)	0.60				
Former smoking	0.12 (−0.37 to 0.60)	0.63				
Dyslipidemia	0.59 (−0.04 to 1.22)	0.06	0.36 (−0.19 to 0.91)	0.20	0.37 (−0.18 to 0.92)	0.19
Ischemic stroke	1.34 (0.07 to 2.61)	0.04	0.81 (−0.28 to 1.90)	0.14	0.91 (−0.20 to 2.02)	0.11
Ischemic heart disease	0.24 (−0.56 to 1.04)	0.56				
Gout	1.27 (0.42 to 2.12)	0.004	0.83 (0.04 to 1.62)	0.04	0.86 (0.06 to 1.66)	0.03
Hypertension	−0.40 (−1.63 to 0.82)	0.51				
Diabetes mellitus	0.50 (−0.12 to 1.12)	0.12				
Atrial fibrillation	−0.57 (−1.60 to 0.46)	0.27				
Dialysis vintage (months)	−0.007 (−0.01 to 0.001)	0.10	0.005 (−0.006 to 0.007)	0.87	0.0009 (−0.006 to 0.007)	0.81
Beta-blocker use	−0.001 (−0.63 to 0.63)	1.00	0.03 (−0.51 to 0.56)	0.92	0.05 (−0.49 to 0.59)	0.85
Statin	0.28 (−0.38 to 0.94)	0.40				
ACEi	−1.80 (−3.38 to −0.22)	0.03	−1.11 (−2.47 to 0.25)	0.11	−1.10 (−2.47 to 0.27)	0.11
ARB	−0.16 (−0.82 to 0.50)	0.63				
Non-calcium-based phosphate binder	0.86 (−0.23 to 1.36)	0.16			0.31 (−0.37 to 1.00)	0.37
Active vitamin D	−0.30 (−0.92 to 0.32)	0.35			−0.16 (−0.74 to 0.41)	0.57
Calcium < 8.5 mg/dL	−0.31 (−1.00 to 0.38)	0.37				
Phosphate						
Normal phosphate	1					
• Low phosphate (<3 mg/dL)	−0.26 (−1.23 to 0.70)	0.60				
• High phosphate (>5 mg/dL)	0.07 (−0.61 to 0.76)	0.83				
PTH						
• PTH 120–585 pg/mL	1					
• PTH < 120 pg/mL	−0.46 (−2.50 to 1.58)	0.66	0.31 (−1.47 to 2.10)	0.73	0.37 (−1.42 to 2.17)	0.68
• PTH > 585 pg/mL	−0.77 (−1.42 to −0.11)	0.02	−0.55 (−1.14 to 0.04)	0.06	−0.48 (−1.11 to 0.15)	0.13
Vitamin D	0.001 (−0.02 to 0.02)	0.93				
Bicarbonate	0.05 (−0.11 to 0.20)	0.56				
ALP > 120 U/L	−1.13 (−1.78 to −0.47)	0.001	−0.52 (−1.14 to 0.10)	0.10	−0.49 (−1.12 to 0.13)	0.12
Albumin < 3.5 g/dL	0.38 (−0.412 to 1.16)	0.34				

Model 1: Adjusted covariates with *p* ≤ 0.1 from the univariate model. Model 2: Model 1 + non-calcium-based phosphate binder and active vitamin D supplementation. Abbreviations: ACEi, angiotensin-converting enzyme inhibitor; ALP, alkaline phosphatase; ARB, angiotensin II receptor blocker; BMI, body mass index; PTH, parathyroid hormone.

**Table 6 medicina-59-00129-t006:** Associated factors with osteoporosis.

Factors	Univariate		Multivariate	
	OR (95% CI)	*p*-Value	aOR (95% CI)	*p*-Value
Age ≥ 65 years	2.47 (1.20–5.06)	0.01	3.31 (1.25–8.80)	0.02
Female	3.66 (1.76–7.62)	0.001	4.13 (1.68–10.14)	0.002
BMI	0.91 (0.84–0.98)	0.02	0.89 (0.81–0.98)	0.02
Alcohol	0.97 (0.64–1.48)	0.91		
Former smoking	0.67 (0.38–1.19)	0.18		
Dyslipidemia	0.59 (0.29–1.21)	0.15		
Ischemic stroke	0.58 (0.13–2.53)	0.47		
Ischemic heart disease	1 (0.41–2.43)	1.00		
Gout	0.53 (0.19–1.45)	0.22		
Hypertension	3.81 (0.76–19.10)	0.10	1.69 (0.76–29.06)	0.10
Diabetes mellitus	1.55 (0.77–3.13)	0.21		
Atrial fibrillation	1.19 (0.38–3.75)	0.77		
Dialysis vintage (months)	1.00 (1.00–1.02)	0.09	1.00 (0.99–1.02)	0.40
Beta-blocker use	1.21 (0.60–2.43)	0.60	1.24 (0.52–2.96)	0.63
Statin	0.75 (0.36–1.58)	0.45		
ACEi	4.20 (0.46–38.66)	0.20		
ARB	1.23 (0.59–2.57)	0.57		
Non-calcium-based phosphate binder	0.43 (0.17–0.10)	0.07	0.54 (0.17–1.73)	0.30
Active vitamin D	1.55 (0.77–3.12)	0.22		
Calcium < 8.5 mg/dL	1.36 (0.62–2.95)	0.43		
Phosphate				
• Normal phosphate	1			
• Low phosphate (<3 mg/dL)	2.63 (0.83–8.32)	0.10	1.68 (0.40–7.10)	0.48
• High phosphate (>5 mg/dL)	0.921 (0.43–1.96)	0.83	1.79 (0.62–5.18)	0.28
PTH				
• PTH 120–585 pg/mL	1			
• PTH < 120 pg/mL	2.55 (0.22–29.31)	0.45	1.11 (0.08–15.63)	0.94
• PTH > 585 pg/mL	1.95 (0.92–4.14)	0.08	1.65 (0.62–4.41)	0.32
Vitamin D	1.00 (0.96–1.01)	0.34		
Bicarbonate	1.02 (0.86–1.22)	0.79		
ALP > 120 U/L	5.06 (2.14–11.96)	<0.001	3.88 (1.33–11.32)	0.01
Albumin < 3.5 g/dL	2.53 (1.00–6.39)	0.05		

Abbreviations: ACEi, angiotensin-converting enzyme inhibitor; ALP, alkaline phosphatase; aOR, adjusted odds ratio; ARB, angiotensin II receptor blocker; BMI, body mass index; OR, odds ratio; PTH, parathyroid hormone.

## Data Availability

Data are available from the corresponding author (P.P.) upon reasonable request.

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
