# Peer review of "The Association of Beta-Blocker Use and Bone Mineral Density Level in Hemodialysis Patients: A Cross-Sectional Study"

_medicina, 2023, doi:10.3390/medicina59010129_

Round 1

Reviewer 1 Report

In the last decade, there have been several studies that showed the beneficial effects of beta-blockers in increasing BMD, including in CKD patients. Therefore, your study focused on HD patients can represent an important step forward for an adequate medical management in this group of population. For a better understanding, it would be nice if you could include a figure presenting the effect of beta-blockers on the reduction of bone resorption and consequently the influence on BMD in CKD patients.

Reviewer 2 Report

The authors have investigated that association between be-ta blocker use and osteoporosis in hemodialysis patient in cross-sectionally study. Although focusing the relationship between beta-blocker treatment and osteoporosis is really interesting in particular to hemodialysis patients, there has several important concerns with research methods.

The authors evaluated osteoporosis by using BMD, but not renal osteodystrophy. The authors need to revise the description in Abstract and Introduction section.

What is the reason that the authors decided the period of be-ta blocker treatment as 20 weeks and how about the dose of be-ta blockers? Generally speaking, I wonder BMD changes so much in 20 weeks?

It seems that the authors found the association between be-ta1 blocker and BMD. So, I think the authors need the background table comparing patients without be-ta1 blocker and with be-ta1 blocker.

In addition, this kind of study will be needed longitudinal evaluation of BMD comparing patients with be-ta1 blocker to patients without.

What is the reason that the authors adapted primary outcome of BMD of femoral neck but not hip, lumbar spine, or radius? Please make these reasons clear.

I think that the factors that are included in the multivariate analyses, should be selected according to the results of previous studies. In particular, it is essential to include “be-ta blocker” and “its confounding factors”. Drug use related CKD-MBD, calcium, inorganic phosphorus, and iPTH levels are common factors that may be associated with the BMD.

(Was PTH intact PTH or whole PTH?)

Although table 2 included the be-ta1 blocker use and non-selective blocker use, is this appropriate for the point of view of Multicollinearity?

Round 2

Reviewer 2 Report

The authors addressed my queries and improved the quality of MS.